# Quantifying the mental health and economic impacts of prospective Universal Basic Income schemes among young people in the UK: a microsimulation modelling study

Tao Chen ,[1,2] Howard Reed ,[3,4] Fiorella Parra-Mujica ,[5] Elliott Aidan Johnson ,[3] Matthew Johnson ,[3] Martin O'Flaherty ,[1] Brendan Collins ,[1] Chris Kypridemos [1]

¹Department of Public Health, Policy & Systems, University of Liverpool, Liverpool, UK
²Center for Health Economics, University of York, York, UK
³Social Work, Education and Community Wellbeing, Northumbria University, Newcastle upon Tyne, UK
⁴Landman Economics Ltd, Colchester, UK
⁵School of Health Policy & Management, Erasmus University Rotterdam, Rotterdam, Netherlands

**Correspondence to**
Chris Kypridemos;
C.Kypridemos@liverpool.ac.uk

## ABSTRACT

**Objective** Universal Basic Income (UBI)—a largely unconditional, regular payment to all adults to support basic needs—has been proposed as a policy to increase the size and security of household incomes and promote mental health. We aimed to quantify its long-term impact on mental health among young people in England.

**Methods** We produced a discrete-time dynamic stochastic microsimulation that models a close-to-reality open cohort of synthetic individuals (2010–2030) based on data from Office for National Statistics and Understanding Society. Three UBI scheme scenarios were simulated: Scheme 1—Starter (per week): £41 per child; £63 per adult over 18 and under 65; £190 per adult aged 65+; Scheme 2—Intermediate (per week): £63 per child; £145 per adult under 65; £190 per adult aged 65+; Scheme 3—Minimum Income Standard level (per week): £95 per child; £230 per adult under 65; £230 per adult aged 65+. We reported cases of anxiety and depression prevented or postponed and cost savings. Estimates are rounded to the second significant digit.

**Results** Scheme 1 could prevent or postpone 200 000 (95% uncertainty interval: 180 000 to 210 000) cases of anxiety and depression from 2010 to 2030. This would increase to 420 000 (400 000 to 440 000) for Scheme 2 and 550 000 (520 000 to 570 000) for Scheme 3. Assuming that 50% of the cases are diagnosed and treated, Scheme 1 could save £330 million (£280 million to £390 million) to National Health Service (NHS) and personal social services (PSS), over the same period, with Scheme 2 (£710 million (£640 million to £790 million)) or Scheme 3 (£930 million (£850 million to £1000 million)) producing more considerable savings. Overall, total cost savings (including NHS, PSS and patients' related costs) would range from £1.5 billion (£1.2 billion to £1.8 billion) for Scheme 1 to £4.2 billion (£3.7 billion to £4.6 billion) for Scheme 3.

**Conclusion** Our modelling suggests that UBI could substantially benefit young people's mental health, producing substantial health-related cost savings.

## INTRODUCTION

There is a crisis in mental health among young people, which will have long-term impacts on well-being and development. Between 1995 and 2014, the proportion of 16–24 years old in the UK reporting a long-standing mental health condition increased from 0.6% to 5.9%.[1] A meta-analysis including 11 high-income countries indicated that one in eight children have mental disorders.[2] Unfortunately, this problem may have been exacerbated since the recent austerity period in the UK[3] and further magnified during the COVID-19 pandemic.[4] Currently, it is estimated that childhood mental disorders are the leading cause of childhood disability globally[5] and incur considerable social and economic burdens to the healthcare system and families.[6 7]

Previous studies have found that adverse economic conditions could negatively affect mental health in children and young people.[8–10] Our previous analysis of Understanding Society data for young people aged

⇒ Mental health and economic potential impacts of Universal Basic Income (UBI) implementations are explored.

⇒ Limited to pre-pandemic data, so impacts of cost-of-living crisis and other changes since the pandemic may change the results.

⇒ It does not include the impact of people moving into higher tax brackets ('fiscal drag').

16–24 from UK households went further, showing a dose-response effect.[11] Young people living in households within the lowest net equivalised income quintile group had a higher probability than the second lowest quintile group of reporting clinically significant symptoms of anxiety and depression; the second lowest had a higher probability than the middle quintile group and so on up the income scale.

To address this public health concern, many approaches have been proposed to promote mental health and prevent mental disorders.[12 13] However, these reactive policies have often focused on individual-level interventions such as improving coping strategies and increasing the efficiency of services. At the same time, interest is growing in addressing the social causes of anxiety and depression. A large body of evidence indicates that social determinants strongly affect those conditions: income, wealth, education, social capital and opportunity.[10 14–16] One proposed means of addressing these issues, which is increasingly gaining support from various organisations, policymakers and politicians, is Universal Basic Income (UBI), a largely unconditional, regular payment to all permanent residents to support basic needs. Johnson and colleagues have set out a theoretical model of its impact that indicates that UBI can mitigate social determinants of health by reducing poverty, mitigating inequality and fostering long-term, health-promoting behaviour.[14 17]

Previous modelling has examined the potential costs and benefits of mental health interventions to prevent or treat anxiety, depression, bipolar disorder and suicide among adolescents[13] or by comparing cognitive behavioural therapy and selective serotonin reuptake inhibitors for major depression in children and adolescents.[18] However, no study assessed the long-term impact of UBI on mental health in children and young people.

This study aimed to quantify the potential impacts of three prospective UBI schemes on the mental health of young people and the associated economic burden during the 2010–2030 period in the UK.

## METHODS
There are multiple pathways for a UBI scheme to impact health. In figure 1, we present a comprehensive model of impact with three distinct but perhaps synergistic biopsychosocial pathways to impact health, including mental health (for more details, please refer to Johnson and colleagues[17]). The present study examines the impact of changes in income, specifically on anxiety and depression. This focuses largely on the pathway associated with poverty reduction. However, the redistributive effects of the schemes modelled may also track the impacts of reduction in inequality. Larger incomes are also often more predictable. The data that informed our models are observational; therefore, it is difficult to disentangle and quantify the pathways in our analysis. Experimental data and qualitative analysis would be required to establish the relative impacts of each pathway.

Our study used two microsimulations in a hybrid serial modelling arrangement to simulate three UBI scheme scenarios and estimate the prevalence of anxiety and depression and consequent deaths under the counterfactual net equivalised household income distributions.

The three UBI scheme scenarios were broadly designed to provide pathways towards attaining the Minimum Income Standard (MIS) with income distributions microsimulated using the Landman Economics Tax-Transfer Model (first microsimulation in the serial arrangement). MIS is the income needed by different types of households to reach a socially acceptable living standard, as determined by members of the public with support from experts.[19] The three schemes are detailed below, and table 1 outlines the cost of each UBI scheme used in the paper showing that they are all fiscally neutral (at least in terms of first-round static effects):

### Scheme 1 – starter (per week): £41 per child; £63 per adult over 18 and under 65; £190 per adult aged 66+
Scheme 1 is a realistic 'starter' scheme with relatively low payments for working age adults and children, but payments for pensioners which are above the level of the current UK state retirement pension for a pensioner with a full record of National Insurance contributions during working life.

### Scheme 2 – intermediate (per week): £63 per child; £145 per adult under 65; £190 per adult aged 66+
Scheme 2 is a mid-point between the Schemes 1 and 3.

### Scheme 3 – MIS level (per week): £95 per child; £230 per adult under 65; £230 per adult aged 66+
Scheme 3 ensures that all families reach the MIS level.

Each of the above schemes is intended to meet the following conditions:
1. UBI would be paid to eligible residents without condition, raising the incomes of the lower income groups.
2. UBI would reduce the percentage gap between the top and bottom income groups through fiscal reform, be high enough to make a material difference in people's lives and raise the level of universality in the social security system, thus reducing reliance on means-testing.
3. UBI would be affordable (although this depends on how this is defined).
4. UBI would minimise losses for low-income households, minimise the amount of disruption involved in moving to a new income support system and enjoy broad public support. For instance, these schemes have been found to enjoy support among critical 'red wall' voters.[20]

### Household income modelling
The Landman Economics Tax-Transfer Model was used with Waves 1–10 (inclusive) of Understanding Society data to microsimulate the UBI payments in Schemes 1, 2 and 3 and corresponding packages of increases to tax (income tax and National Insurance contributions) required to achieve fiscal balance for each scheme, taking into account reductions in payments of existing

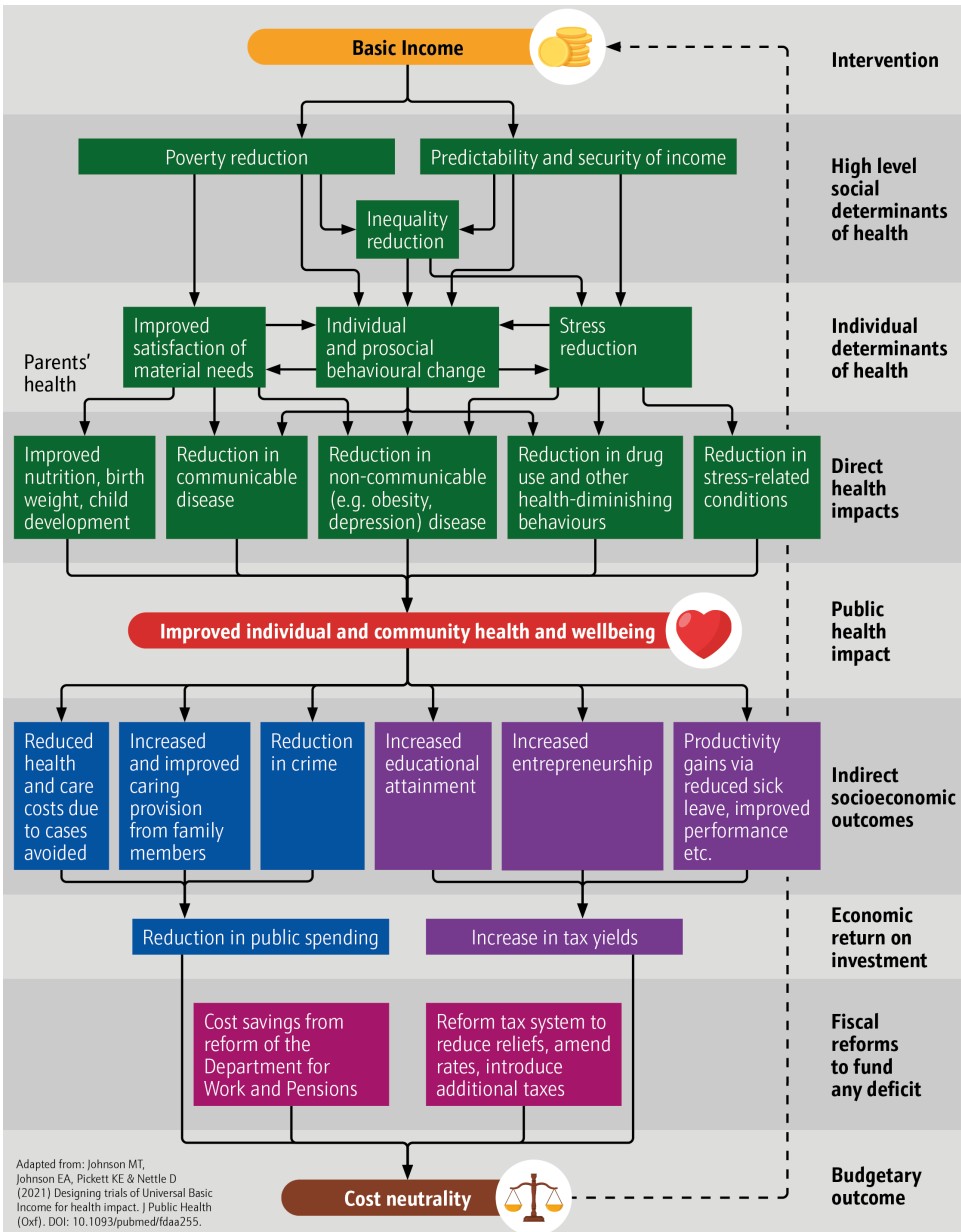

**Figure 1** Universal Basic Income pathways to health and potential cost neutrality. Adapted from Johnson and colleagues.[17]

benefits.[21] The main non-means-tested benefits in the UK benefits system (Child Benefit and the State Pension) are replaced by the UBI in each scheme. UBI payments are counted as unearned income to calculate Universal Credit (a means-tested transfer payment for people on low incomes in the UK benefits system). So UBI payments replace Universal Credit payments one-for-one for low-income individuals (although a small disregard is applied for Schemes 1 and 2 so that low-income individuals and families are better off under UBI than the baseline system. In Scheme 3, UBI payments are sufficiently high that no disregard is necessary). Note that fiscal balancing for each scheme (ie, ensuring that the increase in tax revenue approximately matches the cost of UBI expenditure, net of any reductions in other benefits) is done statically in the model, not taking account

of any behavioural changes in response to the receipt of UBI payments.

The level of payments in each of the schemes is based on existing analysis by Reed and colleagues[21] using data from the UK Family Resources Survey (FRS) for the 2019/2020 fiscal year. This overlaps with the interview dates for Understanding Society Wave 10, so the same level of payments is used for Wave 10 as for the FRS analysis. For earlier waves of Understanding Society, the UBI payments are deflated using the UK Consumer Prices Index. The income tax and National Insurance increases are adjusted in each wave to ensure approximate fiscal balance and to compensate for any change in real incomes between waves of Understanding Society. Therefore, to ensure that the combination of the introduction of UBI and changes to the benefits system, and the accompanying income

**Table 1** Fiscal costings of the three Universal Basic Income (UBI) scheme scenarios (£ billions)

| All costs/savings in £ billions | Scheme 1 | Scheme 2 | Scheme 3 |
|---|---|---|---|
| Gross cost of UBI | 274.4 | 464.3 | 677.5 |
| Benefit savings: | | | |
| Abolition of child benefit | 10.6 | 10.6 | 10.6 |
| Abolition of state pension | 96.9 | 96.9 | 96.9 |
| Reduction in Universal Credit/legacy benefits | 7.1 | 37.4 | 59.6 |
| Total savings | 114.6 | 144.9 | 167.1 |
| Tax changes: | | | |
| Reduction of personal allowance to £750 | 90.9 | 90.9 | 90.9 |
| National Insurance changes | 54.4 | 78.2 | 78.2 |
| Income tax rate increases | 14.7 | 306.9 | 497.7 |
| Total tax increases | 160.0 | 319.7 | 510.5 |
| Net cost | 0.2 | 0.3 | 0.1 |

tax and National Insurance contribution increases, are approximately fiscally neutral in each wave.

## Health and disease costs impact modelling

The second microsimulation in the serial arrangement was a discrete-time dynamic stochastic microsimulation that used the output of the first microsimulation (changes in the distribution of the equivalised household income) and translated into changes in the prevalence of anxiety and depression and the consecutive disease costs. Specifically, the second microsimulation models a close-to-reality open cohort of synthetic individuals (starting at 90 000) representing individuals aged 14–24 in the UK between 2010 and 2030. Their rates of fertility, mortality and migration were driven by Office for National Statistics (ONS) estimates and projections.[22] At the same time, ethnicity, whether born in the UK, highest educational attainment and marital status were informed by waves 1–10 of the Understanding Society: the UK Household Longitudinal Study.[23] We simulated the prevalence of anxiety and depression using the SF-12 Mental Component Summary (MCS) measure with a clinical threshold score of ≤45.6. Our simulation was based on all attributes above, including equivalised household income, based on evidence from Parra-Mujica and colleagues,[11] assuming a causal relationship between income and anxiety and depression and total risk reversibility. For all the attributes described above, we fitted logistic regression models to the Understanding Society data and predicted from them to allocate the attributes of the synthetic individuals. We further assumed that the observed increasing trend of anxiety and depression prevalence would plateau after 2019 to avoid an unrealistic increase over time. We did the same for the equivalised household income trends post 2019.

So, when a synthetic individual enters the simulation, their age and sex are defined based on ONS estimates. Then their ethnicity (white/other), place of birth (UK/elsewhere), education and marital status are estimated using regression models fitted in Understanding Society. Age, sex and year are predictors in all these regressions. Additionally, place of birth was a predictor for ethnicity; ethnicity and place of birth were predictors for education, and all previous attributes were predictors for marital status. Finally, using the Landman Economics Tax-Transfer Model, the household equivalised income is estimated based on all previously simulated attributes and the scenario (baseline or one of the UBI Schemes). For every simulated year, the age of synthetic individuals increases by one and education, marital status and household equivalised income attributes are updated. The prevalence of anxiety and depression is estimated for each synthetic individual based on all previous attributes using a logistic regression fitted in Understanding Society data.

In our microsimulation, we also modelled reductions in cause-excess deaths based on relative risks (RRs) identified in observational data from Denmark by Meier and colleagues.[24] To account for the fact that our case definition of anxiety and depression, based on a self-reported measure (SF-12 MCS), might also include less severe cases compared with the clinically diagnosed cases in the study by Meier and colleagues and reflect the uncertainty of this parameter, we formed a beta-PERT distribution[25] based on estimates from the study mentioned above. We used the low 95% CI of the fully adjusted all-cause mortality RR (1.56) reported for individuals diagnosed with anxiety disorders as the minimum for our beta-PERT distribution, the anxiety disorders RR for ages <30 (4.4) as the mode of the distribution and the dual diagnosis of anxiety and depression RR for ages <30 (6.9) as the maximum of the distribution. These estimates were comparable to published estimates from Sweden, although using slightly different case definitions.[26]

## Costs

Costs for anxiety and depression treatment were informed by the usual care arm of the CADET randomised control

trial.[27] We estimated and report two different cost perspectives: (1) the UK National Health Service (NHS) and personal social services (PSS) (third party payer) perspective, and (2) A broader perspective that included resource use from primary/community care (eg, general practitioner, mental health worker, social worker), secondary care (eg, hospital admissions, psychiatric rehabilitation ward, outpatient appointment), social care (eg, daycare centre, drop in a club), informal care from friends/relatives (eg, hours per week help from friends/relatives), patient other costs (eg, over-the-counter medications, travel costs) to estimate the total cost of anxiety and depression. We inflated all costs to mid-2015 British pounds using the Consumer Price Index and did not apply an annual discount rate for costs occurring in the past or future. Finally, we used the reported mean and SD of the costs to form Gamma distributions using the method of moments and capture the uncertainty of the inputs. We further assumed that only half of the synthetic individuals that reported symptoms of anxiety or depression would seek treatment and thus incur healthcare costs. This assumption was roughly informed by the Adult Psychiatric Morbidity Survey.[28]

### Uncertainty and sensitivity analysis

In all reported figures, we ensured that we captured the uncertainty of the outputs. The microsimulation used a second-order Monte Carlo with 200 iterations for the outer loop and ~90 000 iterations for the inner loop to propagate the uncertainty of the inputs to the outputs.[29] We summarised the uncertainty of the outputs by reporting the median and 95% uncertainty interval (UI) of their respective distributions. The four main sources of uncertainty in the model are: (1) the strength of the relation between equivalised household income and anxiety and depression, (2) the excess mortality risk from anxiety and depression, (3) the disease costs and (4) the individual heterogeneity from the modelled attributes of the synthetic individuals.

The outputs from the model are case-years of anxiety and depression prevented or postponed, deaths prevented or postponed, disease costs from the NHS and PSS perspective and total disease costs. All our estimates are rounded to the second significant digit.

### Patient and public involvement
None.

## RESULTS

The UK population between 14 and 24 years old was projected to increase from approximately 9 million in 2010 to 9.6 million in 2030. Of those, about 2 million would experience symptoms of anxiety and depression in 2010; this was projected to more than double to 4.1 million by 2030. Anxiety and depression were more prevalent in women, about 46%, versus about 30% in men, and their prevalence increased with age. It was also slightly more prevalent among non-white and those born in the UK.

The model estimated that approximately 200 000 (95% UI: 180 000 to 210 000) cases of anxiety and depression could be prevented or postponed in Scheme 1 from 2010 to 2030. The effectiveness would increase to 420 000 (95% UI: 400 000 to 440 000) for Scheme 2 and 550,000 (95% UI: 520 000 to 570 000) for Scheme 3. In relative terms, these represent approximately 0.028% (95% UI: 0.026% to 0.030%) of all case years with anxiety and depression for Scheme 1, 0.059% (95% UI: 0.056% to 0.063%) for Scheme 2 and 0.077% (95% UI: 0.074% to 0.081%) for Scheme 3. Correspondingly, 110 (95% UI: 0 to 430), 320 (95% UI: 0 to 640) and 420 (95% UI: 100 to 770) deaths would be prevented or postponed for the three Schemes, respectively.

Table 2 shows the NHS and PSS cost savings and total cost savings. Overall, the total cost saving, including NHS, PSS and patient-related costs, would range from £1.5 billion (£1.2 billion to £1.8 billion) for Scheme 1 to £4.2 billion (£3.7 billion to £4.6 billion) for Scheme 3.

## DISCUSSION

Our findings indicate the mental health impact that UBI could have on a specific age group through a pathway of increased income. Despite the limited scope of the present modelling study, it is clear that the potential is substantial and significant. Over 21 years, 200 000–550 000 cases of anxiety and depression could be prevented or postponed, saving £330 million to £930 million in health and social services costs. In reality, these are opportunity costs rather than cashable savings; most NHS costs are fixed staffing costs. Since demand typically outstrips supply for NHS mental health services—the prevented cases of anxiety and depression will mean that other people will benefit by receiving treatment more quickly.[30]

To our knowledge, this is the first study to model the health and disease cost impact of UBI among young

**Table 2** Modelling results estimating disease cost savings from different perspectives

| Schemes | National Health Service and personal social services cost savings over 2010–2030, assuming 50% of cases diagnosed and treated | Total cost savings over 2010–2030, assuming 50% of cases diagnosed and treated |
|---|---|---|
| Scheme 1 | £330 million (£280 million to £390 million) | £1.5 billion (£1.2 billion to £1.8 billion) |
| Scheme 2 | £710 million (£640 million to £790 million) | £3.2 billion (£2.8 billion to £3.6 billion) |
| Scheme 3 | £930 million (£850 million to £1000 million) | £4.2 billion (£3.7 billion to £4.6 billion) |

people. Previous modelling mainly focused on assessing mental health prevention through trial-based economic evaluation[18 27] but was subject to inadequate patient follow-up and not capturing the final health outcomes. However, our model-based design is fundamental in an economic evaluation of mental health prevention due to its advantages, including the ability to consider all relevant prospective policy alternatives, including evidence not often collected in trials, and extrapolate beyond the usually short-term horizon of empirical studies.[31]

Our modelling exercise assumes that low income is causally related to anxiety and depression and that increasing income can fully reverse the risk. The association between income and mental health has been shown in experimental and observational studies.[10 15] However, the heterogeneity of cash transfer schemes and other policies intended to redistribute income and reported mental health outcomes makes evidence synthesis difficult. Large, representative trials of UBI that capture comprehensive and comparable data in the real world are crucial.[14]

In the future, it is imperative to develop models that comprehensively capture the health impact of income changes across the entire population and all major disease types. These models should incorporate quality of life measures such as quality-adjusted life years gained, and their value could be evaluated based on National Institute for Health and Care Excellence or UK Treasury valuations. Such an approach would enable assessment of the potential cost savings that could be achieved through improved health outcomes under a UBI policy. Moreover, the additional equity and well-being benefits of UBI, which are not fully captured through a 'burden of disease' perspective, may further offset some of the financial burdens associated with implementing such policies.

Regarding policy implications, the present study provides evidence that UBI can produce health benefits for young people over a medium-term time horizon. This is useful evidence for the basic income trial currently underway in Wales, in which care leavers are offered a basic income of £1600 per month (higher than Scheme 3 in the present study).[32] Care leavers have rates of mental health problems that are up to six times that of the non-care exposed population,[33] so there is potential for a basic income for care leavers to have a greater relative effect in this group, depending on how much mental health problems are related to income in this population. There is also a basic income pilot in Santa Clara County (California, USA), where people leaving foster care at age 25 receive US$1000 a month. Including common outcome measures such as SF-12 in these real-world pilots would provide further data to compare with the results of this study and enable further microsimulation modelling.

The concept of a guaranteed minimum income the state provides to all permanent residents is gaining traction across the political spectrum. Even the conservative-leaning UK think tank Bright Blue recently called for 'the establishment of a new 'minimum living' income',[34]

although largely within the UK welfare system as it is currently constituted. A scoping review of the public health effects of interventions resembling basic income found 'modest to strong positive effects on several health outcomes, including low birth weight, infant obesity, adult and child mental health, service use, and nutrition'.[35] An evidence synthesis based on several studies of basic income programmes found that, overall, basic income improved mental health, with mediating factors being increased free time, hope for the future, and reduced stigma.[36]

Our modelling exercise has some limitations. First, when setting the UBI payment levels and the income tax thresholds in the reform schemes, we assumed that both are consumer price index-uprated between Understanding Society Waves 1 and 10. This means that UBI payments for each adult and child remain constant in real terms from year to year. We made the same assumption about tax. However, this fails to account for the fact that real earnings grew in most years between 2010 and 2019, resulting in a process known as 'fiscal drag' (taxpayers tending to move into higher marginal rate brackets) that would gradually decrease the impact of the UBI schemes. To minimise this bias and considering the turbulent period since 2020, we did not model trends in equivalised household income post 2019. Second, all the data we used were from the years before the COVID-19 pandemic and the post-pandemic cost-of-living crisis. Therefore, the trends we modelled may not be indicative of the post-pandemic period up to 2030. Specifically, the pandemic may increase the prevalence of anxiety and depression in the population and further limit access to appropriate treatments and support. Furthermore, the post-pandemic cost-of-living crisis and the high-inflation period may compress family incomes and accelerate the mental health crisis. These limitations make our modelled estimates conservative and research on UBI policies more relevant than ever.

However, there are also potential sources of bias whose direction and magnitude are unclear. For instance, the modelled effect size of UBI schemes is based on observational data (Understanding Society) that may suffer from selection bias, misclassification, survivorship bias and reverse causality. Although, a recent meta-analysis found that the effect of income changes on mental health was reported as larger when experimental studies were exclusively considered versus when only observational studies were considered.[10] Finally, our modelling does not include wider potentially unintended consequences that the restructuring of the income redistribution system might cause to the economy.

## Conclusions

In summary, the present study suggests that UBI could substantially improve mental health in young people, reduce costs related to the NHS, PSS and patients and reduce premature mortality. These findings add to the

growing body of evidence supporting the potential for UBI to improve population health.

**Contributors** MJ and EAJ conceptualised the study. TC and FPM analysed the data. HR developed and performed the Universal Basic Income modelling. CK developed and performed the health modelling and supervised all the analysis. TC produced the first draft. All authors (MJ, EAJ, TC, FPM, HR, CK, MO, BC) interpreted the results and critically revised the manuscript. CK is the guarantor of the study.

**Funding** This research was funded by a Wellcome Trust Discretionary Award (223553/Z/21/Z): 'Assessing the prospective impacts of Universal Basic Income on anxiety and depression among 14–24-year-olds'. The funder had no role in study design, data collection and analysis, decision to publish or preparation of the manuscript.

**Competing interests** BC works part-time for Welsh Government; this does not represent any views of the Welsh Government. All other coauthors declare no conflict of interest.

**Patient and public involvement** Patients and/or the public were not involved in the design, or conduct, or reporting, or dissemination plans of this research.

**Patient consent for publication** Not applicable.

**Provenance and peer review** Not commissioned; externally peer reviewed.

**Data availability statement** This is a modelling study that generated no new empirical data. Understanding Society is available through the UK data service (https://ukdataservice.ac.uk/).

**ORCID iDs**
Tao Chen http://orcid.org/0000-0002-5489-6450
Howard Reed http://orcid.org/0000-0003-4577-1178
Fiorella Parra-Mujica http://orcid.org/0000-0002-0974-7528
Elliott Aidan Johnson http://orcid.org/0000-0002-0937-6894
Matthew Johnson http://orcid.org/0000-0002-9987-7050
Martin O'Flaherty http://orcid.org/0000-0001-8944-4131
Brendan Collins http://orcid.org/0000-0002-3023-8189
Chris Kypridemos http://orcid.org/0000-0002-0746-9229

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
