## [Reviewer comments · BMJ Open]

ARTICLE DETAILS

TITLE (PROVISIONAL)	Quantifying the mental health and economic impacts of prospective Universal Basic Income schemes among young people in the UK: a microsimulation modelling study
AUTHORS	Chen, Tao; Reed, Howard; Mujica, Fiorella Parra; Johnson, Elliott; Johnson, Matthew; O'Flaherty, M; Collins, Brendan; Kypridemos, Christodoulos

VERSION 1 – REVIEW

REVIEWER	McCartney, Gerry University of Glasgow, Place and Wellbeing Directorate
REVIEW RETURNED	07-Jun-2023

GENERAL COMMENTS	Thank you for the opportunity to review your paper which seeks to model the mental health impacts of different basic income schemes on young people in England. This is an important and interesting study. I have a number of comments on your paper which I have listed below. I hope these are helpful in redrafting. • Abstract: the headline estimates are clearly rounded – it would be helpful to make that explicit in the methods.• NHS cost savings – more nuanced discussion is required about the extent to which that this is actually a real cash saving or simply a notional saving given the large amount of unmet need for healthcare currently in the system and the fixed costs of NHS service provision that make the realisation of real cost savings difficult (see https://doi.org/10.1016/j.puhe.2019.02.013 for example).• Methods – I think it would be helpful to lay out a clear theory of change for your intervention and model, not least to clarify which pathways are being modelled (I think this is restricted to only the additional income and change in some benefits from the UBI schemes) and are not being modelled (e.g. income security pathway, the potential for people to be more active in their community/volunteering, the reduction in low wage/poor quality work, changes in taxes to fund the schemes, etc.).• Methods – you describe Scheme 1 as being fiscally neutral, but without explaining how. At face value, it represents higher payments to some groups without explicit reductions for others or new/changed taxes. It would be very helpful to provide tables/histograms to be explicit about the fiscal costs/spend of each scheme, and the changes in benefits and taxes for each. Depending on the perspective on fiscal and monetary policy adopted, the implications for taxes, other areas of spending,
--

	inflation, etc., there may be other unintended/unmodelled impacts from the policy changes that need to be considered (at the very least as important sources of uncertainty). For example, in Scottish economic modelling of UBI the tax changes required to meet MIS were very large, and these will have health impacts.  • In the modelling you describe using the mental health prevalence at different incomes to ascribe what the change in mental health will be. As you acknowledge, this is an assumption, but there are better sources for estimating the change in mental health now available, at least for working-age adults (see https://www.thelancet.com/journals/lanpub/article/PIIS2468-2667(22)00058-5/fulltext for a meta-analysis of the actual mental health impacts of changes in income). Ideally you would incorporate the effect size from this meta-analysis rather than use the current assumption – not least because many/most of your cohort will move into this age group over the follow-up period. If not this assumption becomes a major limitation of the estimates which needs to be explicitly discussed in the abstract and key messages. • Related to the above, can you clarify how the changing mental health of your cohort is modelled at baseline (i.e. without the interventions), and whether any assumptions are employed? • It would be helpful in the methods to provide a table with all the key modelling assumptions employed, ideally with some narrative on their certainty and any likely direction/scale of bias that they may generate. • In the discussion it would be helpful to outline the sampling limitations and biases, and measurement biases, inherent in Understanding Society.
--	--

REVIEWER	Hamilton, Leah Appalachian State University
REVIEW RETURNED	04-Jul-2023

GENERAL COMMENTS	I have completed a detailed review of your manuscript, "Quantifying the mental health and economic impacts of prospective Universal Basic Income schemes among young people in the UK: a microsimulation modelling study." Overall, your work provides a valuable contribution to the current body of research exploring the potential impacts of Universal Basic Income on mental health. The microsimulation modeling approach, originality, clarity, and potential policy implications of your study are commendable. I hope that the more detailed feedback below will be beneficial in refining and strengthening your manuscript. Kind regards ** Overall feedback: The manuscript is effectively structured and communicates its purpose clearly: to quantify the potential long-term impact of Universal Basic Income (UBI) on the mental health of young
--

	people in England and associated cost savings. It presents original research, and the authors' approach is commendable, employing a discrete-time dynamic stochastic microsimulation model. This innovative approach contributes significantly to the field by providing a comprehensive assessment of UBI's impacts on mental health, which has been previously unexplored in this depth. The authors' approach of modeling three different UBI scenarios and their impact on mental health and cost savings provides a nuanced perspective on potential policy implementations. The simulation-based approach is appropriate for the research question at hand, and the authors demonstrate a deep understanding of the complexities and limitations inherent in this type of modeling. The article is original, presenting a new perspective on the potential impact of UBI schemes on young people's mental health and cost savings in health and social services. It adds to the growing body of research exploring UBI's potential impacts on health outcomes, providing fresh insights into a topic of significant social and policy relevance. The manuscript's language is clear, and no major revisions are needed for readability or comprehensibility. It's written in a scientific manner appropriate for an academic article and should be easily understood by professionals or academics in the field. However, a minor point is to correct a typographical error in the range of cost savings for Scheme 1 in the Abstract and Results sections. Overall, the manuscript is a well-conducted and clearly articulated piece of original research. The use of microsimulation to model the impact of UBI on mental health outcomes and cost savings represents a significant contribution to the field, and the findings offer important implications for policy development and future research. Abstract: The values for health cost savings for Scheme 1 seem to have a typographical error as £280m - £290m would not be a suitable range for the £330m quoted. It might be worth re-checking this data. Otherwise, the abstract stands alone effectively, providing a comprehensive understanding of the study without including unnecessary information. Introduction: The introduction effectively summarizes the topic's current state, painting a vivid picture of the mental health crisis among young people and the potential socioeconomic factors contributing to it. It thoroughly addresses the limitations of current knowledge, particularly the lack of long-term impact studies on Universal Basic Income (UBI) on mental health among children and young people, making the necessity of this study clear. The aim of the study, to quantify the potential impacts of three prospective UBI schemes on young people's mental health and the associated economic burden in the UK, is clearly defined and aligns with the content in the abstract, indicating consistency throughout the manuscript. The research question is clear and appropriate and directly
--	--

	addresses a significant gap in the current knowledge, reinforcing the importance and relevance of this study in the field. Methods: Overall, this is a well-written methods section that successfully conveys the complex procedures and modeling techniques used in this study. Providing a simplified overview of the modeling process would make it more accessible to readers less familiar with these techniques. The section appears to fulfill the necessary criteria for a sound scientific methods section, contributing to the overall validity and integrity of the study. Results: The results section effectively communicates the potential impacts of the three UBI schemes. The study's findings are presented in a clear, quantified manner, allowing for a clear understanding of the projected outcomes for each scheme. The use of uncertainty intervals also gives an understanding of the possible range of outcomes. As in the abstract, it would be beneficial to address the potential typo in the cost savings range for Scheme 1 to ensure accuracy. In the context of the research question, these results appear to be relevant and meaningful, demonstrating the potential benefits of UBI on mental health outcomes and associated economic burdens. Discussion: The discussion and conclusion of the study are robust, providing a clear and logical interpretation of the findings. The authors appropriately align their results with current research, highlighting their unique contribution in modeling the health and cost impacts of Universal Basic Income (UBI). They demonstrate foresight in suggesting future research directions and in addressing the need for a broader approach in evaluating UBI impacts. The conclusions drawn are well-supported by the data, adding to the credibility of the study. Importantly, the authors openly discuss the study's limitations, showing a comprehensive understanding of their assumptions' implications and potential biases. They do not shy away from acknowledging the potential impact of these limitations on their findings, which enhances the study's transparency and credibility.
--	---

VERSION 1 – AUTHOR RESPONSE

Reviewer: 1

Dr. Gerry McCartney, University of Glasgow, Public Health Scotland Glasgow Office

Comments to the Author:

Dear Authors

Thank you for the opportunity to review your paper which seeks to model the mental health impacts of different basic income schemes on young people in England. This is an important and interesting study. I have a number of comments on your paper which I have listed below. I hope these are helpful in redrafting.

Many thanks for your kind and encouraging words.

1. Abstract: the headline estimates are clearly rounded – it would be helpful to make that explicit in the methods.

Thank you, these were rounded to the 2nd significant digit, indeed. We now state this in the last sentence of the methods in the abstract and the main text.

2. NHS cost savings – more nuanced discussion is required about the extent to which that this is actually a real cash saving or simply a notional saving given the large amount of unmet need for healthcare currently in the system and the fixed costs of NHS service provision that make the realisation of real cost savings difficult (see <https://doi.org/10.1016/j.puhe.2019.02.013> for example).

Thank you, this is an important point- in reality, the savings are more like opportunity costs rather than cashable savings to the NHS, but the prevented mental health demand will likely allow other people to be treated more quickly. We have added the underlined sentences to the first paragraph of the discussion section.

“Over 21 years, 200 to 550 thousand cases of anxiety and depression could be prevented or postponed, saving £330m to £930m in health and social services costs. In reality, these are opportunity costs rather than cashable savings; the majority of NHS costs are staffing costs which are fixed. Since demand typically outstrips supply for NHS mental health services –the prevented cases of anxiety and depression will mean that other people will benefit by receiving treatment more quickly.”

3. Methods – I think it would be helpful to lay out a clear theory of change for your intervention and model, not least to clarify which pathways are being modelled (I think this is restricted to only the additional income and change in some benefits from the UBI schemes) and are not being modelled (e.g. income security pathway, the potential for people to be more active in their community/volunteering, the reduction in low wage/poor quality work, changes in taxes to fund the schemes, etc.).

Thank you for raising this critical point. We have added a new figure with the UBI health impact pathways and discuss the implications in the first paragraph of the methods. Our analysis focuses largely on the pathway associated with poverty reduction. However, the redistributive effects of the schemes modelled may also track the impacts of reduction in inequality. Larger incomes are also often more predictable. Because the data are observational, it is difficult to disentangle the pathways in our analysis. Experimental data and qualitative analysis are required to establish the relative impacts of each pathway.

4. Methods – you describe Scheme 1 as being fiscally neutral, but without explaining how. At face value, it represents higher payments to some groups without explicit reductions for others or new/changed taxes. It would be very helpful to provide tables/histograms to be

explicit about the fiscal costs/spend of each scheme, and the changes in benefits and taxes for each. Depending on the perspective on fiscal and monetary policy adopted, the implications for taxes, other areas of spending, inflation, etc., there may be other unintended/unmodelled impacts from the policy changes that need to be considered (at the very least as important sources of uncertainty). For example, in Scottish economic modelling of UBI the tax changes required to meet MIS were very large, and these will have health impacts.

Thank you, we have added a new table (Table 1) in the methods section with the detailed costing of the three modelled UBI schemes. It shows that they are all fiscally neutral, at least in terms of first-round static effects. However, we agree that this is perhaps too simplistic; therefore, we added the sentence below as one of our limitations.

“...our modelling does not include wider potentially unintended consequences that the restructuring of the income redistribution system might cause to the economy.”

5. In the modelling you describe using the mental health prevalence at different incomes to ascribe what the change in mental health will be. As you acknowledge, this is an assumption, but there are better sources for estimating the change in mental health now available, at least for working-age adults (see [https://www.thelancet.com/journals/lanpub/article/PIIS2468-2667\(22\)00058-5/fulltext](https://www.thelancet.com/journals/lanpub/article/PIIS2468-2667(22)00058-5/fulltext) for a meta-analysis of the actual mental health impacts of changes in income). Ideally you would incorporate the effect size from this meta-analysis rather than use the current assumption – not least because many/most of your cohort will move into this age group over the follow-up period. If not this assumption becomes a major limitation of the estimates which needs to be explicitly discussed in the abstract and key messages.

Thank you. We are familiar with the meta-analysis by Thomson et al., which we cite in our paper. We have considered using it instead of our current approach; however, this would have introduced different biases in our estimates because 1) there is very high heterogeneity among the included studies in terms of the exposure, the outcome and the setting from the inclusion of studies in LMIC, despite the meta-regressions that were used, 2) the case definition of ‘mental health’ or actually the lack of it is not suitable for simulation, 3) the definition of the intervention/exposure that was meta-analysed is also blurry, 4) most studies are observational in the main analysis 4) the effect is for an age group that was much wider (16-64) than our focus (14-24), and we argue that age especially for the age group we were interested may modify the effect size 5) The risk of bias for many of the included studies was substantial. Because of these, we couldn’t reliably extract a mathematical relation between a change in income in pounds and a change in anxiety/depression score. For the same reason, we cannot compare the effect size we used with the published one by Thomson et al. That said, we used this meta-analysis (the part restricted to RCTs) and the one by Romero et al. as supporting evidence that indicates a potentially causal relationship between income and mental health and the potential for risk reversibility. Therefore, we used Understanding Society only to quantify the relationship between income and anxiety/depression, which allowed us to calculate this effect more granularly by age, sex, education level, ethnicity (white/other), country of birth (UK/other), and marital status. We argue that our current approach is more transparent about the potential biases that may

entail. We make these biases clearer now at the bullet points immediately after the abstract and in the now expanded limitations paragraph.

“The modelled effect size of the universal basic income is based on observational data (Understanding Society) that may suffer from selection bias, misclassification, survivorship bias, and reverse causality.”

6. Related to the above, can you clarify how the changing mental health of your cohort is modelled at baseline (i.e. without the interventions), and whether any assumptions are employed?

Thank you, we have now added a new paragraph in our methods to describe this more transparently. The paragraph reads:

“So when a synthetic individual enters the simulation, their age and sex are defined based on ONS estimates. Then their ethnicity (white/other), place of birth (UK/elsewhere), education, and marital status are estimated using regression models fitted in Understanding Society. Age, sex, and year are predictors in all these regressions. Additionally, place of birth was a predictor for ethnicity; ethnicity and place of birth were predictors for education, and all previous attributes were predictors for marital status. Finally, using the Landman Economics Tax Transfer Model, the household equivalised income is estimated based on all previously simulated attributes and the scenario (baseline or one of the UBI Schemes). For every simulated year, the age of synthetic individuals increases by one and education, marital status and household equivalised income attributes are updated. The prevalence of anxiety and depression is estimated for each synthetic individual based on all previous attributes using a logistic regression fitted in Understanding Society data.”

7. It would be helpful in the methods to provide a table with all the key modelling assumptions employed, ideally with some narrative on their certainty and any likely direction/scale of bias that they may generate.

Thank you. We have now substantially changed and expanded our bullet points at the beginning of the text, that BMJ Open requires, instead of a table to make our study's key assumptions and limitations more explicit and immediately accessible. We have also expanded our methods and limitations sections to discuss them further, as described in our previous replies and based on your invaluable insights.

8. In the discussion it would be helpful to outline the sampling limitations and biases, and measurement biases, inherent in Understanding Society.

Thank you, we now added the sentence to our limitations

“The modelled effect size of the universal basic income is based on observational data (Understanding Society) that may suffer from selection bias, misclassification, survivorship bias, and reverse causality.”

Reviewer: 2

Dr. Leah Hamilton, Appalachian State University

Comments to the Author:

Dear Authors,

I have completed a detailed review of your manuscript, "Quantifying the mental health and economic impacts of prospective Universal Basic Income schemes among young people in the UK: a microsimulation modelling study."

Overall, your work provides a valuable contribution to the current body of research exploring the potential impacts of Universal Basic Income on mental health. The microsimulation modeling approach, originality, clarity, and potential policy implications of your study are commendable.

I hope that the more detailed feedback below will be beneficial in refining and strengthening your manuscript.

Kind regards

**

Overall feedback:

1. The manuscript is effectively structured and communicates its purpose clearly: to quantify the potential long-term impact of Universal Basic Income (UBI) on the mental health of young people in England and associated cost savings. It presents original research, and the authors' approach is commendable, employing a discrete-time dynamic stochastic microsimulation model. This innovative approach contributes significantly to the field by providing a comprehensive assessment of UBI's impacts on mental health, which has been previously unexplored in this depth.

The authors' approach of modeling three different UBI scenarios and their impact on mental health and cost savings provides a nuanced perspective on potential policy implementations. The simulation-based approach is appropriate for the research question at hand, and the authors demonstrate a deep understanding of the complexities and limitations inherent in this type of modeling.

The article is original, presenting a new perspective on the potential impact of UBI schemes on young people's mental health and cost savings in health and social services. It adds to the growing body of research exploring UBI's potential impacts on health outcomes, providing fresh insights into a topic of significant social and policy relevance.

The manuscript's language is clear, and no major revisions are needed for readability or comprehensibility. It's written in a scientific manner appropriate for an academic article and

should be easily understood by professionals or academics in the field. However, a minor point is to correct a typographical error in the range of cost savings for Scheme 1 in the Abstract and Results sections.

Overall, the manuscript is a well-conducted and clearly articulated piece of original research. The use of microsimulation to model the impact of UBI on mental health outcomes and cost savings represents a significant contribution to the field, and the findings offer important implications for policy development and future research.

Many thanks for your kind words.

Abstract:

2. The values for health cost savings for Scheme 1 seem to have a typographical error as £280m - £290m would not be a suitable range for the £330m quoted. It might be worth re-checking this data. Otherwise, the abstract stands alone effectively, providing a comprehensive understanding of the study without including unnecessary information.

Thank you. This was indeed a typo. We now corrected it and it reads "Scheme 1 could save £330m (£280m - £390m)..."

Introduction:

3. The introduction effectively summarizes the topic's current state, painting a vivid picture of the mental health crisis among young people and the potential socioeconomic factors contributing to it. It thoroughly addresses the limitations of current knowledge, particularly the lack of long-term impact studies on Universal Basic Income (UBI) on mental health among children and young people, making the necessity of this study clear. The aim of the study, to quantify the potential impacts of three prospective UBI schemes on young people's mental health and the associated economic burden in the UK, is clearly defined and aligns with the content in the abstract, indicating consistency throughout the manuscript. The research question is clear and appropriate and directly addresses a significant gap in the current knowledge, reinforcing the importance and relevance of this study in the field.

Methods:

4. Overall, this is a well-written methods section that successfully conveys the complex procedures and modeling techniques used in this study. Providing a simplified overview of the modeling process would make it more accessible to readers less familiar with these techniques. The section appears to fulfill the necessary criteria for a sound scientific methods section, contributing to the overall validity and integrity of the study.

Thank you

Results:

5. The results section effectively communicates the potential impacts of the three UBI

schemes. The study's findings are presented in a clear, quantified manner, allowing for a clear understanding of the projected outcomes for each scheme. The use of uncertainty intervals also gives an understanding of the possible range of outcomes. As in the abstract, it would be beneficial to address the potential typo in the cost savings range for Scheme 1 to ensure accuracy. In the context of the research question, these results appear to be relevant and meaningful, demonstrating the potential benefits of UBI on mental health outcomes and associated economic burdens.

Thank you, we corrected the typo in the results as well.

Discussion:

The discussion and conclusion of the study are robust, providing a clear and logical interpretation of the findings. The authors appropriately align their results with current research, highlighting their unique contribution in modeling the health and cost impacts of Universal Basic Income (UBI). They demonstrate foresight in suggesting future research directions and in addressing the need for a broader approach in evaluating UBI impacts. The conclusions drawn are well-supported by the data, adding to the credibility of the study. Importantly, the authors openly discuss the study's limitations, showing a comprehensive understanding of their assumptions' implications and potential biases. They do not shy away from acknowledging the potential impact of these limitations on their findings, which enhances the study's transparency and credibility.

Thank you.

VERSION 2 – REVIEW

REVIEWER	McCartney, Gerry University of Glasgow, Place and Wellbeing Directorate
REVIEW RETURNED	14-Sep-2023

GENERAL COMMENTS	Dear Authors, Thank you for revising your paper and justifying where you have and have not made changes. Best wishes, Gerry
--

VERSION 2 – AUTHOR RESPONSE